# A Superhydrophobic Alkali Activated Materials Coating by Facile Preparation

Yao Qin, Zhou Fang, Xinrui Chai and Xuemin Cui *

School of Chemistry and Chemical Engineering and Guangxi Key Lab of Petrochemical Resource Processing and Process Intensification Technology, Guangxi University, Nanning 530004, China; 1914302048@st.gxu.edu.cn (Y.Q.); 1914302010@st.gxu.edu.cn (Z.F.); chaixinrui@st.gxu.edu.cn (X.C.)

* Correspondence: cui-xm@mail.tsinghua.edu.cn

**Abstract:** Alkali activated materials (AAMs) were considered as economical and environmentally friendly that have attracted incrementally attention as green coating materials. However, alkali activated materials were inclined to be infiltrated and ruined by harmful ions in water due to their hydrophilicity. And the ordinary ways of construct superhydrophobic coatings were costly, complex and need fluorine material. The superhydrophobic surfaces were fragile owing to the superhydrophobicity of materials were controlled to surface merely. In this work, a facile, convenient and economical strategy to synthesize alkali activated slag materials (AAS) superhydrophobic coatings with excellent water repellence was developed. Herein, the hydrolysis and polymerization of triethoxy (octyl)silane (TTOS) were applied for generating micro/nanostructures to construct a three-dimensional overall superhydrophobic alkali activated slag materials coating. The water contact angle (CA) about surfaces and bottoms of superhydrophobic alkali activated slag materials coatings were 150.2°, 152° and the water rolling angle (SA) of surfaces and bottoms were 5°, 4° respectively. Besides, the superhydrophobic alkali activated slag materials coatings demonstrated excellent mechanical abrasion effect that still maintain super-hydrophobicity after sandpaper abrasion stand. Super-hydrophobicity of coatings could be regenerated by simple sandpaper rubbing when they were attacked chemically. Concisely, the superhydrophobic alkali activated slag materials coatings were show the benefit of affordable and feasibility so that they have the potential for expandable industrial promotion.

**Keywords:** alkali activated slag materials; superhydrophobic; coating; stability





## 1. Introduction

In 1996, Onda [1] proposed the term of "superhydrophobicity" to describe the abnormally grand water contact angle that has not been observed on the surface of hydrophobic materials. The superhydrophobic was defined that the contact angle (CA) is greater than 150° and aka roll angles was not exceeding 10° [2]. There were two noteworthy aspects closely related to the synthesis of superhydrophobic coating [3–5]: the construction of micro/nano surface roughness and modification with low surface free energy materials. Virous techniques have been proposed for constructing surface roughness, including high-temperature induction [6], electrodeposition [7], electroplating [8], chemical etching [9] and sol-gel [10]. A variety of low surface energy materials have been discovered, mainly polymer [11], organo-silane [9,10,12–14], and fatty acid materials [15]. Nonetheless, the conventional procedure for preparing superhydrophobic coatings were usually complex and costly.

Alkali-activated materials (AAMs) is an inorganic polymer produced by mixing aluminosilicate precursors (e.g., granulated blast furnace slag, metakaolin, fly ash) with alkaline activators (e.g., water glass, sodium hydroxide, potassium hydroxide) [16,17]. Noorina [18] proved that sintered kaolin-GGBS geopolymer that underwent the curing process at the

temperature of 60 °C featured the high strength value of 8.90 MPa. Lv [19] prepared a geopolymer-based drying powder decorative coating that that has excellent properties of high strength, artificial aging resistance, high temperature resistance, good workability, etc. Given all of that, alkali activated materials display outstanding acid resistance, sulfate resistance, heat resistance, low permeability and creep, high early strength and environmental friendliness properties as a results of special three-dimensional network structure [18,19]. Owing to the excellent nature, alkali activated materials were widely used as film-forming substances in inorganic water-based coatings [19–21], those coatings were named alkali activated materials-based inorganic coatings [22]. Lv [23] reported a novel strategy of a protective coating for reinforced concrete by alkali-activated slag by compositing potassium methyl silicate, which decreased the corrosion rate of reinforced rebar by 97%. Nicoleta [24] prepared and tested fireproof inorganic coatings based on sodium silicate solution with intumescent additions, which sensibly reduced the rate of temperature increase (up to 75%) in the steel substrate. Salar Lashkari [25] found AAMs have shown better performance compared to cement-based mortars by evaluating the durability characteristics and mechanical properties of AAMs and cement-based coating mortars in the harsh environment of the Persian Gulf. Thus, the utilize of alkali activated materials coatings would minimize the accumulation of solid waste, those were defined as a sustainable material depend on economical and environmentally friendly [20–22]. To date, alkali activated materials coatings have been widely researched and applied like architectural coatings, marine anti-corrosion coatings, heat-resistant anti-corrosion coatings, inorganic metal heat treatment protective coatings, inorganic lubricating wear-resistant coatings, and wood coatings [26]. However, the alkali activated materials are super-hydrophilic due to they take abundant hydroxyl groups, so harmful ions such as sulfate and chloride ions in water can destroy the structure along with water intrusion [26,27]. Superhydrophobic modification can elevate the impermeability and durability of coatings via preventing water intrusion [28]. Qing [29] used calcium stearate as modification for AAS cement to reduce water absorption rate. So far, most hydrophobic modification agents used for alkali activated materials are alkyl silane or silane derivatives [28], but the traditional modification method is not only restricted by the usage of multitudinous of silane that resulting in low economic benefit but it is also limited to the surface [29]. Consequently, their hydrophobic stability and mechanical performance are often inadequate [30]. When the surface is destroyed by stripping and rubbing, the hydrophobic effect is fall [31,32].

Herein, we propose a facile impregnation process for superhydrophobic modification to prepare an AAS coating with overall superhydrophobic properties. Compared with traditional methods of constructing superhydrophobic AAS, the improved method can not only greatly reduce the dosage of silane but also simplify the preparation process, which is a solution to the difficulty of reducing cost in industrial applications. Besides, the mechanism of triethoxyoctylsilane (TTOS) modified AAS was summarized through microscopic analysis. Especially, the prepared superhydrophobic coating has superlative wear resistance, and the CA of the coating increases gradually with the lengthen of wear distance. The work does not require additional reagents and is expected to introduce an effective way to prepare durable, environmentally friendly and practical superhydrophobic AAS coatings. This study provides a new idea for the superhydrophobic modification of AAS and the practical application of superhydrophobic alkali activated slag materials.

## 2. Materials and Methods

### 2.1. Materials

Ground granulated blast furnace slag (GGBS) with irregular sheets and a mean size of approximately 10 μm were provided by Beihai Chengde Steel Company. The chemical compositions of slag were CaO, 43.40%; $SiO_2$, 27.28%; $Al_2O_3$, 3.22%; MgO, 9.95%; $SO_3$, 1.75%; $TiO_2$, 1.17%; $K_2O$, 0.60%; $Na_2O$, 0.54%; $Fe_2O_3$, 0.29%; MnO, 0.13% and 0.01% LOI, which were determined by X-ray fluorescence (XRF). TTOS ($C_{14}H_{32}O_3Si$ > 97%, AR) were applied by Macklin Biochemical Technology Co., Ltd. (Shanghai, China). Dry powder

water glass (modulus M = 2.8 (SiO$_2$/K$_2$O molar ratio)) were produced by the Zhongfa water Glass Factory. (Foshan, China). Q235 steel (Fe > 99%) and glass slides were acquired from Guantai Metallic Materials Co., Ltd. and Fangyuan Technology Co., LTD. (Guangdong, China). Quartz sands (200 mesh) were provided by Ming Hai Environmental Protection company. Deionized waters were prepared in the laboratory. All the raw materials were used directly without any further purification.

### 2.2. Fabrication of Superhydrophobic Alkali Activated Materials Coating

The procedure for preparing the superhydrophobic alkali activated materials coating is shown in Figure 1. The experiment procedure includes three steps.

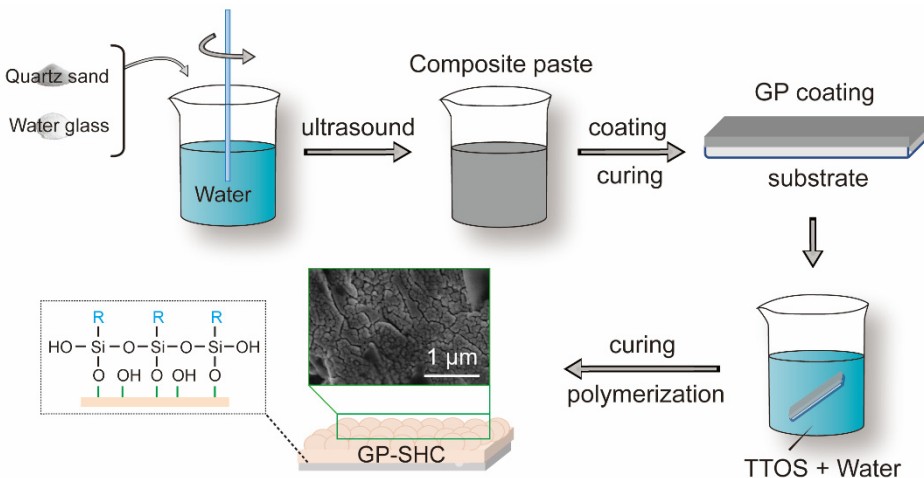

**Figure 1.** Schematic diagram of the preparation of GP-SHC.

Firstly, the preparation method of the original alkali activated materials (AAS) coatings referred to our previous works. Quartz sands were addicted to improve the cracking resistance and roughness of the coating. The mass ratio of slag, dry water glass, quartz sand was 10:3:20. Then, the composite alkali activated material slurry was prepared by adding deionized water into a plastic beaker containing the original slurry to adjust the liquid to solid ratio of the slurry to 35 wt.%, and then stirring the slurry at a speed of 800 r/min for 5 min in a high-speed magnetic agitator. The fresh paste was coated on the Q235 steel substrate or the glass plate by a spatula and then cured under a room temperature of 25 °C and 80% humidity without sealed conditions. The substrate was cleaned with acetone and polished with sandpaper before use. The obtained original AAMs coatings were referred to as GPs.

Then the TTOS mixture composition (TTOS: deionized = water was 1.6 g:165 g) was ultrasonic for 1 h to prepare the modified fluid.

Finally, the abovementioned GPs were immersed in modified composition fluid for 1 h at room temperature of 25 °C and 80% humidity and then dried at same condition, those superhydrophobic AAMs coatings named GP-SHCs.

### 2.3. Characterizations

A contact angle meter (KRUSS, Germany, drop volume is approximately 3 μL) was used to determine the hydrophobicity and super-hydrophobicity of the resultant coatings. The Hitachi SU822 (Tokyo, Japan) field emission scanning electron microscope (FESEM) system was used to analyze the structure of the coating at an accelerated voltage of 10.0 kV. X-ray diffraction (XRD) analysis was performed using Rigaku MinFlex 600 instrument (Tokyo, Japan) with Cu Kα radiation (λ = 0.15419 nm) over 2θ ranging from 5 to 90° and a step length of 0.02°. The chemical structure of the coating was measured by Fourier transform infrared spectroscopy (FT-IR IRTracer-100, Tokyo, Japan and FT-IR Nicolet iS50,

Tokyo, Japan). The elemental states of the coatings were determined by using X-ray photoelectron spectroscopy (XPS, Thermo ESCALAB 250XI).

The robustness test by abrasion resistance that was taking 100-grit sandpapers used as the abrasive material. The detailed test strategy was that coatings were loaded with a 200-g stainless steel weight and were pushed 20 cm under the pushing force of 1 m/min. One cycle was coatings were pushed 20 cm for 50 times. The water resistance exam of GP-SHC coatings were that samples were immersed in artificial seawater for 14 day (d). The chemical stability of GP-SHCs were measured by immerse the coatings into hydrochloric acid solution of pH = 1 and sodium hydroxide solution of pH = 14 for 48 h. Then their reproducibility of superhydrophobic was satisfied by rubbing with sandpaper. All experiments and test have been repeated for many times.

## 3. Results and Discussion

### 3.1. Wettability

The non-wettability characteristics of the coatings were assessed by images of water droplets, CA and SA. Images of water droplets on the GP's surface, GP-SHC's surface and GP-SHC's bottom are shown in Figure 2a–c. The GPs were hydrophilic, its' surface containing many hydroxyl groups, that make water droplets fall into the coating quickly, and the CA is only 16.8° after water droplets touching the surface for 15 s (Figure 2a). As expressed in Figure 2b,c,e,f, it was surprising to note an extreme increase in CA values after TTOS modification. The CA of GP-SHC 's surface and bottom were 150.2°, 156° and the SA of GP-SHC 's surface and bottom were merely 5° and 4° respectively, those achieve the goal of superhydrophobic standard. In addition, the shape of water droplets of actual photographs on the surfaces of the GP-SHC exposed regular spheres. Similarly, the shape of water droplets of actual photographs on the bottom of GP-SHC also present regular spheres but the sphere was more perfectly as shown in Figure 2d,f. Therefore, it confirms that the GP-SHC coatings displayed awesome super-hydrophobicity and super-hydrophobicity not restrict in surface like conventional superhydrophobic coatings.

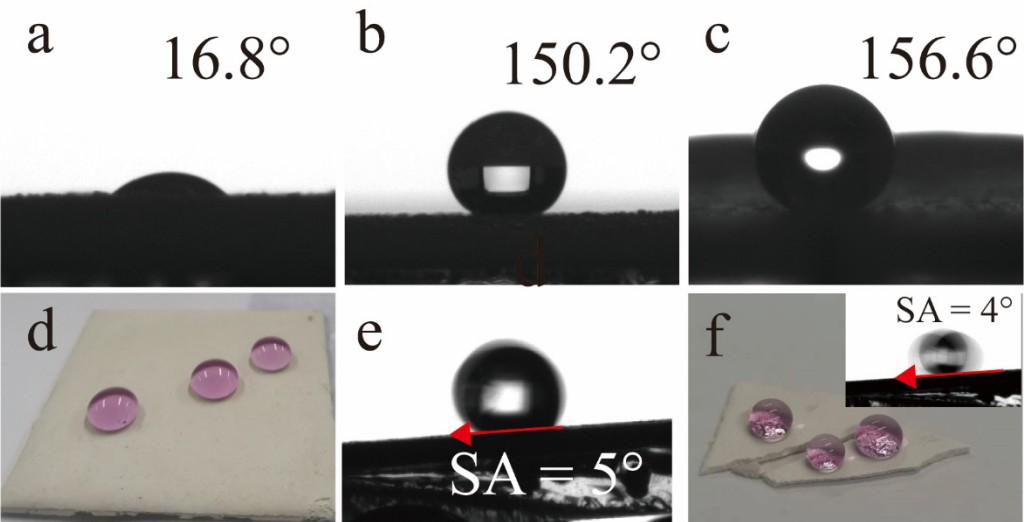

**Figure 2.** Coating contact angle: (**a**) surface of GP, (**b**) surface of GP-SHC, (**c**) bottom of GP-SHC; (**d**) water droplet contact diagram of GP-SHC's surface, (**e**) roll angle of GP-SHC's surface; (**f**) water droplet contact diagram of GP-SHC's bottom.

### 3.2. Surface Structure Characterization

In this paper, the original AAMs that named GP was an inorganic polymer produced by mixing and reacting aluminosilicate precursors of GGBS particles and alkaline activators of water glass. When these particles were mixed with water, alkaline-source particles first dissolved to form an alkaline solution. Then, GGBS were leached into molecules, and the alkali activator broke Si-O and Al-O bonds and releases monomers, gradually forming

four-coordinated $[AlO_4]^-$ and $[SiO_4]^-$, after which it grows into a gel phase [18,19]. Finally, the gel was eventually converted to a solid coating after nucleation and condensation.

The surface microstructures of the GP were investigated by SEM, as proved in Figure 3a–c. The GP film was smooth, compact and featureless without any protrusions. The GP-SHC surfaces could generate various protrusions with about 1μm and connected with to each other as seen in the Figure 3d,e. Figure 3f presents that the strips with uniform shape and size on the surface when further enlargement of the protrusions. The GP-SHC surface exhibited excellent superhydrophobic properties due to the double-layer rough structure. The cross-section and the bottom structure of GP-SHCs were also observed in order to verified GP-SHCs acted overall super-hydrophobicity. It was appeared that the cross-section also had similar protrusions but protuberant structures were more obvious arrangement. More or less, the shape and size of the objects arranged on the protrusion structure were relatively regular spheres that slightly contrast with facial protrusion structure as behold in Figure 3g,h. And the existences of bottom nanostructures were similar to the body of the surface nanostructures, but it led to more closely oriented nanostructures (Figure 3i) his result is consistent with the above finding that the CA of GP -SHC bottoms were only slightly increased according to surfaces (Figure 2b,c).

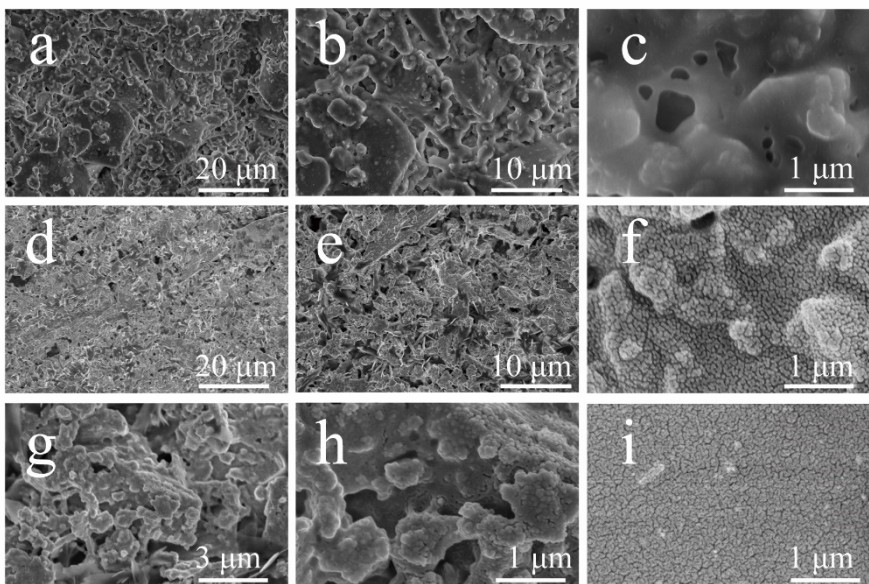

**Figure 3.** FE-SEM microstructure of the coating about different part: (**a–c**) surface morphology of GPs; (**d–f**) surface morphology of GP-SHCs; (**g,h**) cross-sectional morphology of GP-SHCs, (**i**) bottom morphology of GP-SHCs.

*3.3. Crystalline Phases and Chemical Composition Analysis*

The crystalline phases of samples were determined by XRD. The peaks indicating quartz from raw materials in accordance with silica's standard card PDF#87-2096 as exhibit in Figure 4a [18]. The position and intensity of the peaks were similar, which indicates that no new crystalline phase was formed in samples with the addition of TTOS.

It must be admitted that low surface energy is an indispensable factor for preparing superhydrophobic coatings. FT-IR was used to analyze the chemical composition, as shown in Figure 4b. GP-SHC exhibited new peaks at 2926 cm$^{-1}$ and 2856 cm$^{-1}$, which were attributed to C-H tensile vibration peaks and were provided by TTOS when compared with the GP and GP-SHC [27]. The results demonstrated that methyl groups successfully adhered to the surface of the alkali activated materials coating after TTOS modification, which reduced the surface energy of the GP-SHC. Additionally, a band located at approximately 1456 cm$^{-1}$ was related to the stretching vibration of O-C-O bonds. The specific frequency is the characteristic of $CO_3^{2-}$ bonds [33]. The intensity of peak at 1456 cm$^{-1}$ and 872 cm$^{-1}$

increased after TTOS modification, it might be due to water ingress into the coating during modification leads to enhanced carbonation [23,34]. The high intensity broad band between 1200 cm$^{-1}$ and 900 cm$^{-1}$ (Figure 4b) corresponds to asymmetric stretching of Si-O-T bonds (where T corresponds to a tetrahedron of Si or Al) [35]. The Si-O-T peaks of the coating shift to higher wavenumbers after TTOS modification, which is attributed to silicate being replaced by aluminate (theoretically, the vibration mode of Si-O is higher than that of Al-O because Si-O-Si is stronger than Si-O-Al and Al-O-Al bonds) [35]. This indicates that TTOS was helpful to the development of a crystallized state because silicate was replaced by aluminate due to the development of a crystalline state at the early stage of alkali activated materials curing.

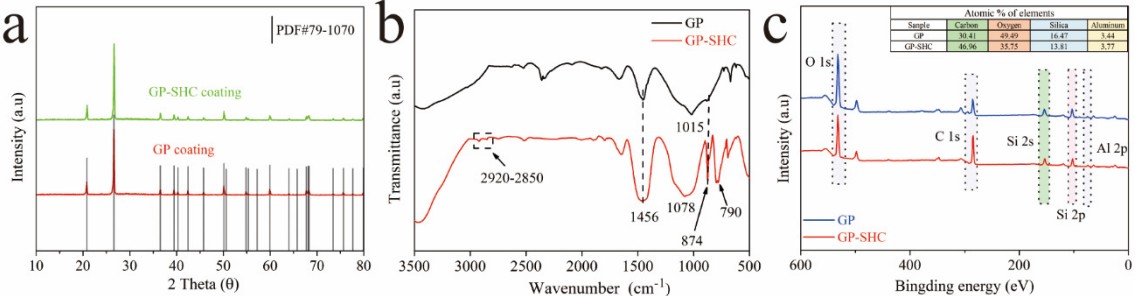

**Figure 4.** (**a**) XRD spectra of the AAS coatings of GP and GP-SHC, (**b**) FT-IR spectra of the AAS coatings of GP and GP-SHC, (**c**) XPS full spectra of the AAS coatings of GP and GP-SHC.

To further explore the formation mechanism of super-hydrophobicity, XPS characterization analysis was carried out. The binding of each element accurately reflects the chemical properties of its constituent subunits. It is vital to determine the reaction by observing the element position changes of carbon, oxygen, silicon and aluminum. As shown in Figure 4c, the major elements such as carbon, silicon, oxygen and aluminum in all coatings are labeled. In the GPs, the content of O atoms is higher than that of C and Si because of the hydrophilic property [36]. After TTOS modification, the C 1s peak was obviously enhanced, and the C atom content of GP-SHC increased from 30.41% to 46.98%, indicating that the content of C was increased. This is an evidence of the attachment of long-chain hydrocarbons to the surface of the AAS coating successfully [37].

Figure 5 shows the high-resolution Si 2p and O 1s spectra of the GP and GP-SHC. The binding energy of Si 2p is mainly composed of the four binding energies of Si-O-H, Si-O-T (T is Al or Si), Si-O-Si and O-Si-C [38]. After modification, the contribution of the Si-O-H binding energy about GP-SHCs decreased, while the combined contribution of Si-O-T and Si-O-Si increased. This result suggests that the surface hydroxyl groups underwent a dehydration condensation reaction with the silanol group formed by the hydrolysis of the silane agent [39]. Furthermore, the O 1s peak of high binding was provided by Si-O-H and the low binding energy peak was attributed to Si-O-Al and Si-O-Si [40]. Similarly, the contribution of the high binding energy peak (Si-O-H) of surfaces decreased while the contribution of the low binding energy peak (Si-O-Al and Si-O-Si) of surfaces increased after coatings were treated with TTOS [41]. The results are consistent with the Si 2p peak, which further confirms that the TTOS molecule and the AAS substrate formed Si-O-Si or Si-O-T groups, resulting in the formation of chemical bonds [42].

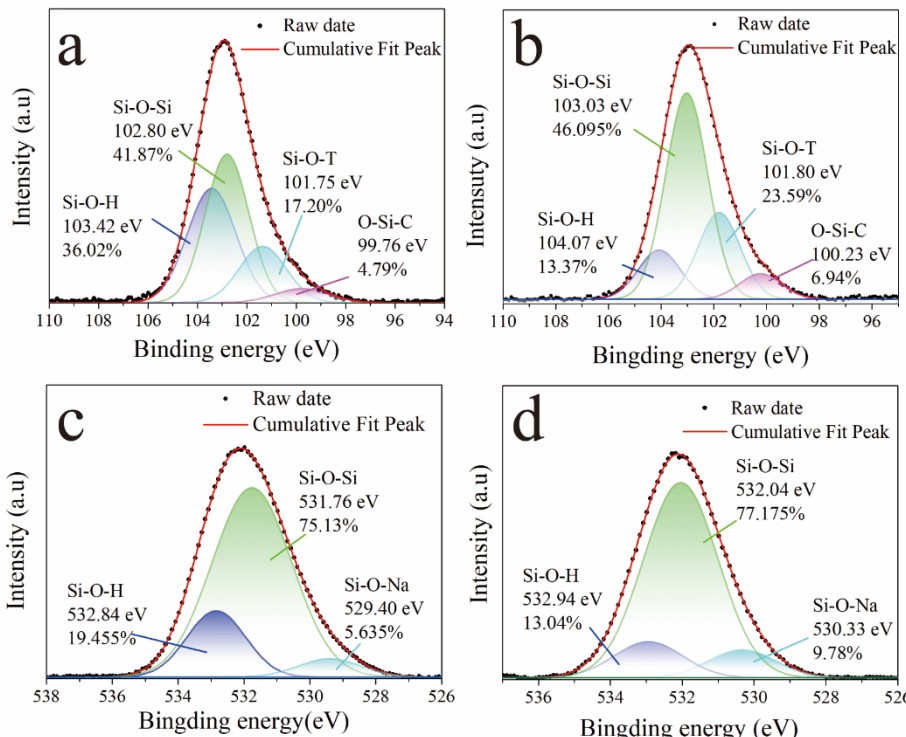

**Figure 5.** XPS narrow scan: (**a**,**b**) Si 2p spectra of GPs and GP-SHCs, (**c**,**d**) O 1s spectra of GPs and GP-SHCs coating.

Based on the above discussion, the mechanism of silane modification of the AAMs is proposed and illustrated in Figure 6 When the TTOS agent is added to water and ultrasonicated, the three ethoxy groups of a TTOS molecule are transformed into hydroxyl groups by the energy provided by ultrasound, which can produce silanol. And at the same time, some self-condensation phenomena may occur to form nanoparticles containing hydroxyl groups, which can a pre-growth effect on nanoparticles [37]. When the coating was soaked in aqueous solution, triethoxyoctyl silanol and deionized water are quickly absorbed into the pores of the coating due to the super hydrophilic effect of the coating. Then, a water-silanol mixed film layer was formed on the surface. When drying, water evaporates and breaks through the silanol coating to form tiny cracks, exposing the hydroxyl groups on the coating surface to form growth points. Moreover, the catalytic action of alkaline substances can be stimulated because the coating is rich in alkaline groups. Finally, the triethoxyoctylsilanol was self-polycondensation and condensation with Si–OH/Al–OH of the alkali activated materials surface form nanostructures. The extension of the carbon chains leads to super-hydrophobicity. For different structures, particles tend to grow dispersed on the substrate due to strong hydrogen bonding. This growth results in a small structure with dense and continuous granular layers (GP-SHC). Substrates with relatively few hydroxyl groups exhibit stronger IM (including hydrogen bonding and van der Waals forces) forces than substrate surfaces. In the presence of a strong IM force, the pre-self-condensing nanoparticles that grow along with the connected particles lead to large coral-shaped particle clusters [38–41].

**Figure 6.** Mechanism of slag-based alkali activated materials coating formation (R = n-octyl).

### *3.4. Abrasion Resistance and Chemical Stability*

The micro/nanostructure and low surface energy were two crucial feature which contribute to the superhydrophobic property. However, these characteristics are highly susceptible to mechanical abrasion, which could decline the water-repellent performance. So, anti-wear performance of the as-prepared superhydrophobic AAS coating was evaluated needful. The schematic illustration of abrasion test employed was shown in Figure 7a, as described in the experimental Section 2.3. The abrasion test results of contact angles presented in Figure 7b,c. It can be seen that the CAs of GP-SHCs were increased to 155.2° after one abrasion cycle as veiled in Figure 7b. After two cycles, part of the coating was completely lost that exposing part of the substrate (Figure 7c,d) but the CA of GP-SHC increased to 156.1°. It was clarity that the CA of GP-SHCs gradually increased with the wear distance became far, which may be ascribable to the increased roughness and the exposure of the interior superhydrophobic structure after wear. The abrasion test confirmed the fabricated GP-SHC act excellent mechanical durability due to the integral coating keep super-hydrophobicity.

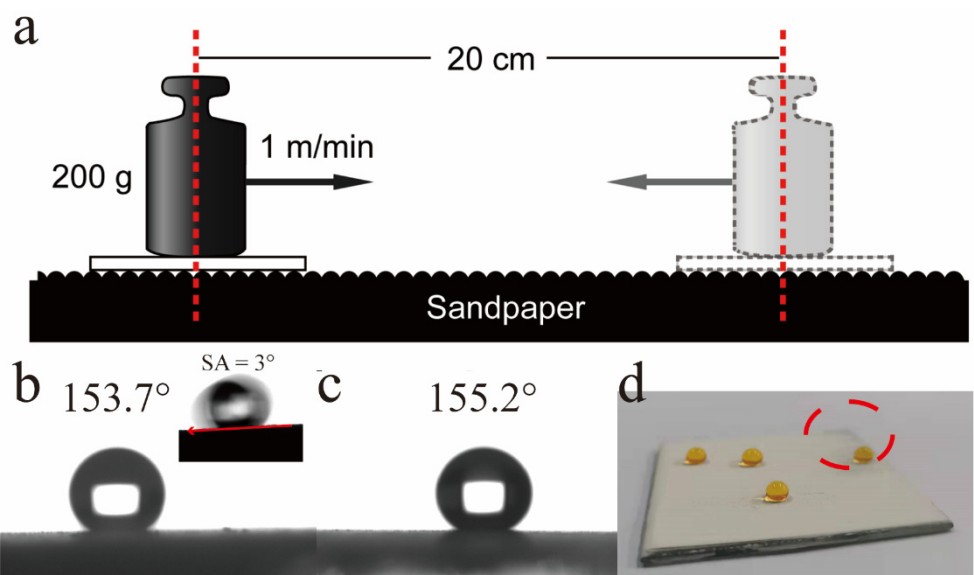

**Figure 7.** The wettability of the GP-SHCs after repeated rubbing: (**a**) the schematic diagram of wear resistance test, (**b**)the CA of GP-SHC surfaces after one cycle, (**c**) the CA of GP-SHC after two cycles, (**d**) the actual photo of GP-SHC after two cycles.

The chemical stability of the GP-SHCs was tested through different solutions, and the regeneration performance about superhydrophobic of the GP-SHCs was explored. The results are shown in Figure 8. The results proved that the GP-SHCs' CA were affect by

artificial seawater hardly that retain 154.13° after coatings were immersed in artificial seawater for 14 d. Moreover, the results professed the GP-SHCs reflected poor corrosion resistance ability to strong acid and alkali solution, but it withstands acid stronger than alkali. Even so, the GP-SHCs remained hydrophobic after being immersed in hydrochloric acid solution of pH = 1 and sodium hydroxide solution of pH = 14 for 48 h and their CAs were drop to 138.99° and 110.14° indivisibly. Notably, the GP-SHCs regained their super-hydrophobicity after simple sandpaper rubbing, which was thanks to exposed new nanostructures by simple sandpaper rubbing when the facial super-hydrophobicity was lost. It is clear from this study that the GP-SHCs are suitable for practical used.

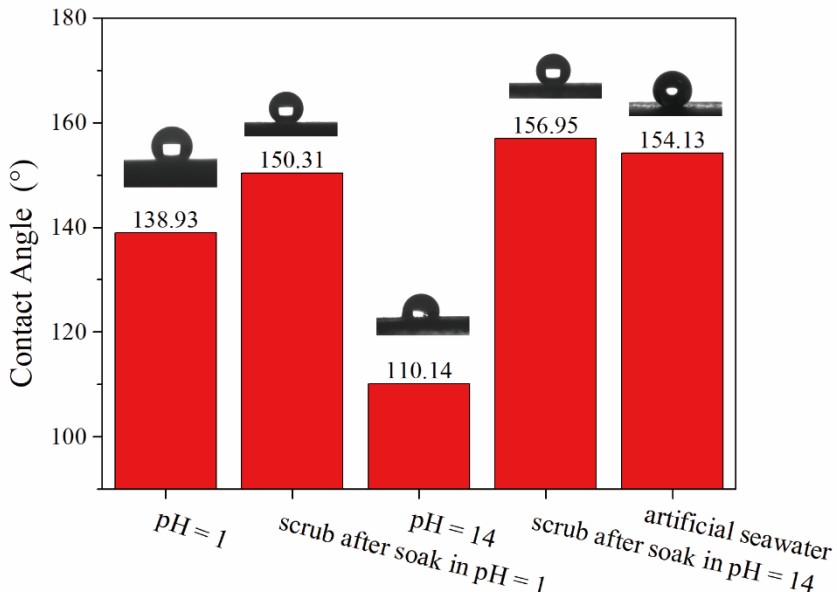

**Figure 8.** Durability study of coatings in different solutions.

## 4. Conclusions

In this experiment, a top-down three-dimensional structured superhydrophobic coatings were prepared through two-phase system (TTOS and water) to modify the alkali-activated slag. The effect of TTOS on the formation of chemical structure of AAS superhydrophobic coatings were developed. The main conclusions are as follows:

(1) TTOS were hydrolyzed and then continued to self-polymerize and copolymerize with the coating surface to grow nanoparticles, which provided excellent superhydrophobic property to coating surface with a contact angle of 150.2° and a roll angle of 5°.

(2) The superhydrophobic coating expressed high stability against external force damage due to the overall hydrophobic modification. The results disclose that superhydrophobic alkali activated materials coatings still keep super-hydrophobicity after they were rubbed with 100-grit sandpaper until the substrate is exposed and were immersed in artificial seawater for a long time. And the coating can be regenerated by simple sandpaper rubbing after being damaged by chemical corrosion.

(3) The work provides new ideas for the superhydrophobic modification of alkali activated materials and novel vote for the practical application of superhydrophobic alkali activated materials.

**Author Contributions:** Y.Q.: Writing—original draft, Software, Methodology, Data curation. Z.F.: Investigation; X.C. (Xinrui Chai): Data curation. X.C. (Xuemin Cui): Conceptualization, funding acquisition, project administration. All authors have read and agreed to the published version of the manuscript.

**Funding:** This work was supported by the Chinese Natural Science Fund (Grant: 51772055) and the Guangxi Natural Science Fund (Grant No.: 2022GXNSFDA035062).

**Institutional Review Board Statement:** Not applicable.

**Informed Consent Statement:** Not applicable.

**Data Availability Statement:** Not applicable.

**Acknowledgments:** This work was supported by the Chinese Natural Science Fund (Grant: 51772055) and the Guangxi Natural Science Fund (Grant No.: 2022GXNSFDA035062).

**Conflicts of Interest:** The authors declare no conflict of interest.

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
