# Peer review of "A Superhydrophobic Alkali Activated Materials Coating by Facile Preparation"

_coatings, doi:10.3390/coatings12060864_

Round 1

Reviewer 1 Report

Dear,

In this research, a facile, convenient and economical strategy to syn thesize geopolymer superhydrophobic coatings with excellent water repellence was developed.  Herein, the hydrolysis and polymerization of triethoxy(octyl)silane (TTOS) were applied for gen erating micro/nanostructures to construct a three-dimensional overall superhydrophobic geopolymer coating.  The research does not require additional reagents and is expected to introduce an effective way to prepare durable, environmentally friendly and practical superhydrophobic geopolymer coatings. The paper is interesting, clearly written and shows that it could have a practical application. Nevertheless, I believe that some additional explanations are needed.

My comments and questions can be found in the pdf version of the manuscript. 

Sincerely

Author Response

Comments and reply
Reviewer #1:

  • Comment (1): Line 89 what did you used Phaeophyta tricornutum for? Explain it?What is it or what is its correct name?

Thank you for the suggestion. I am sorry about that was my oversight. The Phaeophyta tricornutum is irrelevant to this paper. And I have deleted the sentence.

  • Comment (2): Line 93, the term virgin is inappropriate in this term.

I would like to express my thanks and approval for the suggestions put forward by the reviewers. I have replaced “virgin” with “original”. (Line 113, page3).

  • Comment (3): Line 97 I don’t understand in which way you brushed the fresh paste.

Thank you for your feedback. In this experiment, a small scraper is used to coat the fresh slurry on the glass substrate. I have modified the relevant sentences as follows (Line 117, Page3). The sentence before (blue, similarly hereinafter) and after the modification was listed below.

The fresh paste was brushed on the Q235 steel substrate and the glass plate then cured at room temperature.

The fresh paste was coated on the Q235 steel substrate or the glass plate by a spatula and then cured under a room temperature of 25℃ and 80 % humidity without sealed conditions.

  • Comment (4): Line 99 For brevity?

I am very grateful to the reviewers for their comments. I have deleted the phrase of “For brevity” without changing the original meaning (Linec122, Page3).(

  • Comment (5): Line101-105 the procedure needs to be described in a more correctly way.

I am very grateful to the reviewers for their comments. I rewrote the procedure without changing the original meaning, and try our best to modify some sentences as follows. (Line 110-128, Page3)

The preparation of the superhydrophobic coating is display in Fig too. 1 First, 1 g TTOS was mixed with 165 g deionized water and hydrolyzed under ultrasound for 1 hour. Then, the abovementioned completely cured GPs were immersed in hydrolytic solution at room temperature for 1 hour. Finally, the samples were dried at room temperature and then named GP-SHCs.

The procedure for preparing the superhydrophobic alkali activated materials coating is shown in Fig. 1. The experiment procedure includes three steps.

Firstly, the preparation method of the original alkali activated materials (AAS) coatings referred to our previous works. Quartz sands were addicted to improve the cracking resistance and roughness of the coating. The mass ratio of slag, dry water glass, quartz sand was 10:3:20. Then, the composite alkali activated material slurry was prepared by adding deionized water into a plastic beaker containing the original slurry to adjust the liquid to solid ratio of the slurry to 35 wt.%, and then stirring the slurry at a speed of 800 r/min for 5 minutes in a high-speed magnetic agitator. The fresh paste was coated on the Q235 steel substrate or the glass plate by a spatula and then cured under a room temperature of 25℃ and 80 % humidity without sealed conditions. The substrate was cleaned with acetone and polished with sandpaper before use. The obtained original AAMs coatings were referred to as GPs.

Then the TTOS mixture composition (TTOS: deionized =water was 1.6g:165g) was ultrasonic for 1 hour to prepare the modified fluid.

Finally, the abovementioned GPs were immersed in modified composition fluid for 1 hour at room temperature of 25℃ and then dried at same condition, those superhydrophobic AAMs coatings named GP-SHCs.

  • Comment (6): Line 107 What was used substrate.

Thank you for your feedback. Glasses or Q235 steel were used as substrates (Line 128, Page3).

  • Comment (7): Line 124-125 Why did you choose test chemical stability of GP-SHCs pH=1 and pH=14? Is this a standard procedure? Do the test condition in which the obtained materials/coatings can be used?

Thank you for your feedback. There is no established standard for the test of chemical stability of superhydrophobic coatings, so the test of chemical stability in this paper refers to the relevant published literature [1,2](Line 148-149,Page 4).

It can be used in the corrosion prevention of steel and the antifouling of marine infrastructure due to coating express excellent super hydrophobicity.

[1] Zhen Hong, Hanwen Jiang, Mingshan Xue, SiC-enhanced polyurethane composite coatings with excellent anti-fouling, mechanical, thermal, chemiacal properties on various substrates, Progress in organic coatings 168 (2022) 106909

[2] Hua Xie, Jinfei Wei, Shuyi Duan,Non-fluorinated and durable photothermal superhydrophobic coatings based on attapulgite nanorods for efficient anti-icing deicing, Chemical Engineering Journal 428(2022) 132585

  • Comment (8): Line unclear sentence

I am very grateful to the reviewers for their comments. I have remoted the sentence as follows (Line 172-174, Page5).

In this paper, a mixture of alkaline-source particles and GGBS particles synthesized GP.

In this paper, the original alkali-activated materials named GP was an inorganic polymer produced by mixing and reacting aluminosilicate precursors of granulated blast furnace slag and alkaline activators of water glass.

  • Comment (9): Line 154 Why eventually the gel was converted to a solid coating?

Thank you for your feedback. Alkali activated material is a novel three-dimensional network gel material with a high degree of polymerization obtained by the depolymerization, polycondensation and solidification of slurry to get high strength. This is the characteristic of alkali activated material(Line178,Page5).

  • Comment (10): Line 175-176 which base did you used to identify the phase composition.

Thank you for you feedback. The crystal phase composition was obtained through the comparative analysis of jade's standard cards (Line 199-201,Page5).

  • Comment (11): Line 208 You need to improved Figure4.The titles on it are not visible.

I am very grateful to the reviewers for their comments. I have improved Figure4 and revised the relevant parts as follows ((Line 232-234, Page6).

Figure 4. (a) XRD analysis, (b) FT-IR analysis, (c) XPS spectra.

Figure 4. (a) XRD spectra of the AAS coatings of GP and GP-SHC, (b) FT-IR spectra of the AAS coatings of GP and GP-SHC, (c) XPS full spectra of the AAS coatings of GP and GP-SHC.

  • Comment (12): Line 236 This sentence is not clear, please explain it.

I am very grateful to the reviewers for their comments. I have explained the reasons for the growth of nanostructures as follows(Line262-264,Page7).

The nanostructures were formed by self-polymerization and polymerization with the geopolymers.

Triethoxyoctylsilane was preliminarily hydrolyzed under ultrasonic energy, and the ethoxyl group breaks, exposing hydroxyl groups. The completely cured alkali activated coating was immersed in the preliminarily hydrolyzed triethoxyoctylsilane water two-phase solution system. Due to the super hydrophilic effect of the coating, triethoxyoctylsilane and deionized water were quickly absorbed into the pores of the coating. Then, the triethoxyoctylsilanol was self-polycondensation and condensation with Si–OH/Al–OH of the alkali activated materials surface to form nanostructures with the evaporation of water and catalysis of alkaline conditions (Line 262, Page7).

  • Comment (13): Line 252-258 This discussion regarding abrasion resistance should be expand.

I am very grateful for the reviewer’s suggestion. I have expanded the discussion regarding abrasion resistance. (Line 279-292, Page8)

Reviewer 2 Report

The manuscript entitled "A superhydrophoboc geopolymer coating by facile preparation" presents an interesting experimental study conducted on the obtaining of coatings made of alkali activated GGBS. However, the number of tested samples isn’t presented and many other issues must be addressed. The paper needs major revisions before it is processed further, some comments follow:

·       Please analyze the literature related to the differences between the alkali activated materials and geopolymers. Considering the raw material used and its chemical composition, the obtained mixture should be considered alkali activated materials, not geopolymers. Please provide comments about these aspects in the manuscript or make corresponding replacements.

·       The introduction section could be improved. The citations have been introduced in bulk form [6,7,8], [9,10,11,12], [20,21,22], [23,24,25] and note discussed separately. Please discuss the highlights individually and assure a clear correspondence between the affirmations from the manuscript and those from the cited papers.

·       Line 40-42, the definition of geopolymers is inaccurate and ambiguous. Multiple types of raw materials have been used to obtain these materials and the terms "ordinarily excited by alkali" are ambiguous.

·       Lines 47-55, multiples affirmation doesn’t have a background in the literature. Please introduce corresponding studies and assure a clear correlation between the citations and the affirmations.

·       Line 69. Does triethoxysilane stand for Triethoxyoctylsilane? Please make corresponding corrections.

·       Line 93. Please replace the term “virgin” with a more suitable one, such as "fresh".

·       Line 94: the mass ratio was 10:3:20. Why only this ratio was considered. How was this parameter established? Based on preliminary results or previous studies?

·       Line 98: "cured at room temperature". What humidity? How have been maintained the samples (sealed/unsealed)?

·       Line 99: What was the roughness of the surface?

·       Figure 3 – Please introduce figure labels to highlight the areas of interest for the reader.

·       XRD analysis. Please provide a higher resolution image. It seems that in GP coating spectra there is a peak around 57 that does not appear in GP-SHC coating. Therefore, the affirmation from line 178 is false. Also, the FTIR spectra show significant differences. Please discuss.

·       Also, please specify how many samples have been tested from each batch.

· Please improve the conclusions and present them following the main recommendations by the Academia of giving the conclusions of the study by points with highlights.

Author Response

Reviewer #2:

  • Comment (1): Please analyze the literature related to the differences between the alkali activated materials and geopolymers. Considering the raw material used and its chemical composition, the obtained mixture should be considered alkali activated materials, not geopolymers. Please provide comments about these aspects in the manuscript or make corresponding replacements.

Thank you for the suggestion. I agree with the reviewer by looking up relevant literature, so I have replaced geopolymer with alkali activated materials.

  • Comment (2): The introduction section could be improved. The citations have been introduced in bulk form [6,7,8], [9,10,11,12], [20,21,22], [23,24,25] and note discussed separately. Please discuss the highlights individually and assure a clear correspondence between the affirmations from the manuscript and those from the cited papers.

I am very grateful to the reviewers for their comments. I have supplemented, discussed and modified relevant references (Line 40-43, Page1,75-83, Page1).

  • Comment (3): Line 40-42, the definition of geopolymers is inaccurate and ambiguous. Multiple types of raw materials have been used to obtain these materials and the terms "ordinarily excited by alkali" are ambiguous.

I am very grateful to the reviewers for their comments. I have revised the relevant parts as follows. (Line 46-48, Page 2)

Geopolymers are a new type of gel material which produced by raw materials, mainly including industrial solid waste of aluminosilicates such as blast furnace slag and steel slag, were ordinarily excited by alkali [13, 14].

Alkali-activated materials (AAMs) is an inorganic polymer produced by mixing aluminosilicate precursors (e.g., granulated blast furnace slag, metakaolin, fly ash) with alkaline activators (e.g., water glass, sodium hydroxide, potassium hydroxide).

  • Comment (4): Lines 47-55, multiples affirmation doesn’t have a background in the literature. Please introduce corresponding studies and assure a clear correlation between the citations and the affirmations.

Thank you for the suggestion. I had added information required of corresponding studies (Line6-72, Page2) and related literature review of background as explained above (Line69, Page2).

  • Comment (5): Line 69. Does triethoxysilane stand for Triethoxyoctylsilane? Please make corresponding corrections.

Thank you for the suggestion. I am awfully sorry for my negligence. I have corrected the relevant contents as follows:

Besides, the mechanism of triethoxysilane (TTOS) modified slag-based geopolymers was summarized through microscopic analysis.

Besides, the mechanism of triethoxyoctylsilane (TTOS) modified slag-based alkali activated materials was summarized through microscopic analysis (Line8, Page2).

  • Comment (6): Line 93. Please replace the term “virgin” with a more suitable one, such as "fresh".

I would like to express my thanks and approval for the suggestions put forward by the reviewers. I have replaced “virgin” with “original”. (Line 113, page3)

  • Comment (7): Line 94: the mass ratio was 10:3:20. Why only this ratio was considered. How was this parameter established? Based on preliminary results or previous studies?

Thank you for your feedback. The preparation method of the original alkali activated materials (AAS) coatings and the parameter referred to our previous works. Quartz sands were addicted to improve the cracking resistance and roughness of the coating (Line116, Page3).

  • Comment (8): Line 98: "cured at room temperature". What humidity? How have been maintained the samples (sealed/unsealed)?

I would like to express my thanks and approval for the suggestions put forward by the reviewers. I have modified the relevant sentences as follows (Line 121,127, Page3).

The fresh paste was brushed on the Q235 steel substrate and the glass plate then cured at room temperature.

The fresh paste coated on the Q235 steel substrate and the glass plate by a spatula then cured under a room temperature of 25℃ and 80 % humidity without sealed conditions.

  • Comment (9): Line 99: What was the roughness of the surface?

Thank you for your feedback. Roughness of the surface refers to the small spacing and small peaks and valleys of the surfaces. The distance (wave distance) between two peaks or troughs is very small (below 1mm). The smaller the surface roughness, the smoother the surface. Surface roughness is an important factor to achieve superhydrophobic property. In superhydrophobic surfaces, air bubbles are caught in the ups and downs positions of the surface, when water is put on the surface, it does not contact with all surface points. Due to low energy of surface and the presence of air, water does not penetrate into the valley, and consequently, the surface area reduces causing a reduction of friction, and droplets slip on the surface easily.

  • Comment (10): Figure 3 – Please introduce figure labels to highlight the areas of interest for the reader.

I am very grateful to the reviewers for their comments. I have revised the relevant parts as follows(Line195-197,Page5).

Figure 3. SEM images: (a, b, c) surface of GPs; GP-SHCs; (d, e, f) surface morphology, (g, h) cross-sectional morphology, (i) bottom.

Figure 3. FE-SEM microstructure of the coating about different part: (a, b, c) surface morphology of GPs; (d, e, f) surface morphology of GP-SHCs; (g, h) cross-sectional morphology of GP-SHCs, (i) bottom morphology of GP-SHCs.

  • Comment (11): XRD analysis. Please provide a higher resolution image. It seems that in GP coating spectra there is a peak around 57 that does not appear in GP-SHC coating. Therefore, the affirmation from line 178 is false. Also, the FTIR spectra show significant differences. Please discuss.
  • Thank you for the I am awfully sorry for my negligence. I have replaced a higher resolution image (Line228, Page 5) and have corrected the relevant contents as follows. I have replaced the XRD analysis chart and the standard reference card with PDF#87-2096. Therefore, the affirmation from line 178 is not false. (Line 199-202, Page 5)
  • Thank you for the I have expanded the analysis of FTIR spectra as follows(Line203-221,Page5):

It must be admitted that low surface energy is an indispensable factor for preparing superhydrophobic coatings. FT-IR was used to analyze the chemical composition, as shown in Fig. 4b. GP-SHC exhibited new peaks at 2926 cm-1 and 2856 cm-1, which were attributed to C-H tensile vibration peaks and were provided by TTOS when compared with the GP and GP-SHC [27]. The results demonstrated that methyl groups successfully adhered to the surface of the alkali activated materials coating after TTOS modification, which reduced the surface energy of the GP-SHC. Additionally, a band located at approximately 1456 cm−1 was related to the stretching vibration of O-C-O bonds. The specific frequency is the characteristic of CO32− bonds [34]. The intensity of peak at 1456 cm−1 and 872cm-1 increased after TTOS modification, it might be due to water ingress into the coating during modification leads to enhanced carbonation [23,35]. The high intensity broad band between 1200 cm-1 and 900 cm-1 (Fig. 4b) corresponds to asymmetric stretching of Si-O-T bonds (where T corresponds to a tetrahedron of Si or Al) [36]. The Si-O-T peaks of the coating shift to higher wavenumbers after TTOS modification, which is attributed to silicate being replaced by aluminate (theoretically, the vibration mode of Si-O is higher than that of Al-O because Si-O-Si is stronger than Si-O-Al and Al-O-Al bonds) [36]. This indicates that TTOS was helpful to the development of a crystallized state because silicate was replaced by aluminate due to the development of a crystalline state at the early stage of alkali activated materials curing.

  • Comment (12): Also, please specify how many samples have been tested from each batch.

Thank you for the suggestion. I'm very sorry about the number of experiments that was didn't mention in the paper. All the experiments and test have been repeated for many times. I had supplemented the sentence in the paper (Line 150, Page4).

  • Comment (13): Please improve the conclusions and present them following the main recommendations by the Academia of giving the conclusions of the study by points with highlights.

Thank you for the suggestion. I rewrote the Abstract without changing the original meaning, and try our best to modify some sentences of the paper(Line 302-326,Page9).

In this experiment, a top-down three-dimensional structured superhydrophobic coatings were prepared and characterized. TTOS, as a modification agent, were hydrolyzed and then continued to self-polymerize and copolymerize with the coating surface to grow nanoparticles. The superhydrophobic coating provided high stability against external force damage, maintaining superhydrophobic properties due to the overall hydrophobic modification. The results disclose that superhydrophobic geopolymer coatings still keep super-hydrophobicity after they were rubbed with 100-grit sandpaper until the substrate is exposed and were immersed in artificial seawater for a long time. And the coating can be regenerated by simple sandpaper rubbing after being damaged by chemical corrosion. The work provides new ideas for the superhydrophobic modification of geopolymers and novel vote for the practical application of superhydrophobic geopolymers.

In this experiment, a top-down three-dimensional structured superhydrophobic coatings were prepared through two-phase system (TTOS and water) to modify the alkali-activated slag (AAMs).  The effect of TTOS on the formation of chemical structure of AAMs superhydrophobic coatings were developed. The main conclusions are as follows:

(1)  TTOS were hydrolyzed and then continued to self-polymerize and copoly-merize with the coating surface to grow nanoparticles, which provided excellent superhydrophobic property to coating surface with a contact angle of150.2â—¦ and a roll angle of 5â—¦.

(2)  The superhydrophobic coating expressed high stability against external force damage due to the overall hydrophobic modification. The results disclose that superhydrophobic alkali activated materials coatings still keep super-hydrophobicity after they were rubbed with 100-grit sandpaper until the substrate is exposed and were immersed in artificial seawater for a long time. And the coating can be regenerated by simple sandpaper rubbing after being damaged by chemical corrosion. The work provides new ideas for the super-hydrophobic modification of alkali activated materials and novel vote for the practical application of superhydrophobic alkali activated materials.

Round 2

Reviewer 2 Report

The author adressed most of my comments and the paper was improved accordingly. 

The paper can be processed further after the following requirements are considered.

Conclusion section

Line 310: Please remove the abbreviation AAS because it is unnecessary for the following lines.

Conclusion 2- Please split it into two parts. "The work provides new ideas...." - this should be conclusion 3.

Author Response

Thank you for the letter and for the reviewers’ further comments concerning our manuscript entitled “A superhydrophobic alkali activated materials coating by facile preparation”. I have further revised the paper according to the reviewers' comments. Revised portion are marked in red and highlight in the paper. The comments are reproduced and our responses are given directly afterward in a different color (red).

Thanks again.

COMMENTS TO THE AUTHOR:

  • Line 310: Please remove the abbreviation AAS because it is unnecessary for the following lines.

Thank you for your suggestion. I am very sorry for my negligence. I had deleted the abbreviation of AAS (Line312, Page 9)

  • Conclusion 2- Please split it into two parts. "The work provides new ideas...." - this should be

Thank you for your suggestion. I had split the conclusion into two parts. (Line325-327, Page10 )